# Multilevel Structural Components Detection and Segmentation toward Computer Vision-Based Bridge Inspection

**DOI:** 10.3390/s22093502

**Published:** 2022-05-04

**Authors:** Weilei Yu, Mayuko Nishio

**Affiliations:** Department of Engineering Mechanics and Energy, University of Tsukuba, 1-1-1 Tennodai, Ibaraki, Tsukuba 305-8577, Japan; nishio@kz.tsukuba.ac.jp

**Keywords:** bridge inspection, computer vision, components segmentation, multilevel detection

## Abstract

Bridge inspection plays a critical role in mitigating the safety risks associated with bridge deterioration and decay. CV (computer vision) technology can facilitate bridge inspection by accurately automating the structural recognition tasks, especially useful in UAV (unmanned aerial vehicles)-assisted bridge inspections. This study proposed a framework for the multilevel inspection of bridges based on CV technology, and provided verification using CNN (convolution neural network) models. Using a long-distance dataset, recognition of the bridge type was performed using the Resnet50 network. The dataset was built using internet image captures of 1200 images of arched bridges, cable-stayed bridges and suspension bridges, and the network was trained and evaluated. A classification accuracy of 96.29% was obtained. The YOLOv3 model was used to recognize bridge components in medium-distance bridge images. A dataset was created from 300 images of girders and piers collected from the internet, and image argumentation techniques and the tuning of model hyperparameters were investigated. A detection accuracy of 93.55% for the girders and 82.64% for the piers was obtained. For close-distance bridge images, segmentation and recognition of bridge components were investigated using the instance segmentation algorithm of the Mask–RCNN model. A dataset containing 800 images of girders and bearings was created, and annotated based on Yokohama City bridge inspection image records data. The trained model showed an accuracy of 90.8% for the bounding box and 87.17% for the segmentation. This study also contributed to research on bridge image acquisition, computer vision model comparison, hyperparameter tuning, and optimization techniques.

## 1. Introduction

As the service life of infrastructure such as bridges increases, it becomes exposed to varying degrees of safety risks from decay and deterioration. To determine and maintain the safety of such infrastructure, there needs to be regular inspection. Bridge inspection is a critical step in bridge maintenance. The traditional approach by expert inspectors, using manual tools and sensors, is highly subjective, with the associated drawbacks of being time-consuming, laborious, and costly. In recent years, with the development of CV technology, UAV can help improve bridge inspection by acquiring image data without traffic obstructions, and by accessing areas difficult to reach by humans. Other methods to achieve the image data acquisition work may require the inspectors to use large equipment or to work facing safety risks [1]. Thus, automation and semi-automation of UAV-based inspection work has been advocated. The most significant part of such automated inspection work is the identification of the captured bridge images to obtain information on bridge type and components, providing the essential information for a more detailed diagnosis. Additionally, the ability to identify structural details of a bridge allows the UAV to determine the direction of its next action, or to help locate its position on the bridge. The application of a range of artificial intelligence technologies, such as deep learning, computer vision, and image processing techniques, is enabling UAV to identify and judge targets automatically.

With this background of limitations in conventional inspection systems, bridge inspection and diagnosis based on artificial intelligence technologies have been widely studied, with applications for nondestructive testing, damage detection and diagnosis, bridge dynamics, and static load evaluations. A suite of intelligent inspection equipment and technologies such as UAVs and robots, and various data science algorithms such as data mining, computer vision, and deep learning enhance bridge inspection techniques and effectively improve their accuracy and efficiency [2,3]. As the core technology of intelligent image recognition, computer vision automatically extracts valuable information from image data to qualitatively or quantitatively understand or represent the physical world. Computer vision methods can be used to automate traditional human vision tasks; it is an application based on deep learning, which will be described in Section 3. The initial efforts to apply computer vision methods started in the 1960s, and attempted to extract information about the shape of objects using edges and primitive shapes [4]. As image patterns were developed, computer vision methods began to consider more complex perceptual problems, including for example, optical character recognition [5], face recognition [6], and pedestrian and vehicle detection, etc. Currently, the main research areas in computer vision are image classification, object detection, and image segmentation [5,6].

Image classification is one of the most standard applications in computer vision, mainly used for face recognition [7] and object–scene recognition [8]. Image classification sorts an image into a single category, usually corresponding to the most prominent object in the image. The large-scale ImageNet Image Recognition Challenge (ILSVRC), which started in 2010, has created a demand for research into CNN-based algorithms for image recognition [9]. Face recognition, which focuses on identity detection based on facial features, is also a widely researched area, and its accuracy has been significantly improved [10]. Currently, CNN models commonly used for image classification include AlexNet, VGG-Net, ZF-Net, and GoogleNet.

Object detection can identify multiple targets in an image, and localize different targets by outputting bounding boxes. This target detection can be applied to intelligent surveillance [11], autonomous driving [12], and security systems. The commonly used objective detection models include fast-RCNN, YOLO, SSD, MobileNet, and ShuffleNet.

Image segmentation can be categorized into semantic segmentation, instance segmentation, and panoramic segmentation. Semantic segmentation classifies each pixel point in an image into corresponding classes, thus achieving pixel-level classification. Instance segmentation requires pixel-level classification and distinguishes different classes of instances. Panoramic segmentation is a generalization of semantic segmentation and instance segmentation. Unlike semantic segmentation, panoramic segmentation needs to distinguish individual object instances. In addition, the targets in panoramic segmentation are required to be non-overlapping. Image segmentation has been applied to many fields, such as medical imaging, pedestrian detection, and traffic control. The commonly used target segmentation models include FCN [13], Deeplab [14], DenseNet [15], and MaskR-CNN [16].

Combining computer vision technology with remote camera and UAV acquisition offers a promising non-contact solution for civil infrastructure assessment. This system can realize the automatic and consistent transformation of images or video data into useful information [11]. Computer vision applications for civil infrastructure are now recognized as critical components for improved inspection and monitoring. Many studies have been carried out on the use of CV techniques for detection and monitoring of civil infrastructure. Spencer and Narazaki et al. [17] reviewed recent advances in CV techniques for civil infrastructure assessment, presenting the applications of computer vision techniques to infrastructure inspection and monitoring. Inspection applications include using images for characterizing structural components, identifying local and overall visible damage, and detecting changes. Monitoring applications include measuring static strain and quantifying static and dynamic displacement for modal analysis. Zhu et al. [18] addressed the influence of subjective or empirical factors in manual inspection, through the use of migration learning and CNN to automatically analyze and identify many bridge inspection images, improving the accuracy and efficiency of detection and identification. Suzuki et al. [19] used CNN to determine the degree of damage to bridge components, and conducted a comparative study of two bridges with multiple classifications of bridge damage based on CNN. Combined with the analysis of questionnaires from manual inspectors, the method was able to effectively identify the degree of damage of bridge components. Liang et al. [20] proposed a deep learning method based on Bayesian optimization to analyze post-disaster inspection images of reinforced concrete bridge systems, and used different convolutional neural networks to achieve an intelligent evaluation of bridge performance at three levels: system fault classification, bridge combination detection, and local damage localization. Dung et al. [21] proposed an inverse learning method based on deep convolutional neural networks. By fine-tuning the trained, fully-connected layer with the top convolutional layer in a VGG16 model, combined with data enhancement, excellent performance of crack detection at steel bridge connections was obtained. Kurisu et al. [22] constructed a CNN model to determine the level of damage to bridge members based on image data acquired from manual bridge inspections. Grad-CAM was applied to verify the CNN detection model, which provided a visual basis for determining the damage level. By comparing the heat map from Grad-CAM, they determined that the features used in the CNN to judge the damage level were consistent with those used by the inspection engineers.

Many previous studies have focused on the detection and evaluation of specific damage by applying CNN-based deep learnings to image data. However, the automation and semi-automation of bridge inspection work should include not only a focus on damage detection, but also the recognition of bridge type and components, in order to provide an efficient approach for UAV bridge inspection systems. In this study, we categorized the recognition tasks involved in automatic bridge inspection into three levels, based on the recognition distance and the applicable computer vision technique. A CNN model for each task was constructed, and its performance and applicability were evaluated.

The content of this paper is as follows: Section 2 describes multilevel bridge detection and component segmentation, and explains the relationship of the related CNN principles to CV techniques. Section 3, Section 4 and Section 5 present our CNN model constructions, and their results are shown for each of the three levels of recognition tasks. Section 3 shows the classification of bridge type using Resnet50 for the far-distance recognition level; the bridge component detection by YOLOv3 for the mid- distance recognition is contained in Section 4; and the segmentation of bridge components by Mask-RCNN for the close-distance recognition is set out in Section 5. Section 6 summarizes the study and discusses future work.

## 2. CV-Based Framework for Multilevel Bridge Inspection

Figure 1 shows the overall structure of a multilevel structural component detection and segmentation model for bridge inspection. Here, there were three levels of detection/inspection; bridge types, bridge components, and structural members. Image data were assumed to be collected by drones from far- to close- distances to the target bridge. The acquired images were processed to form the data sets for training corresponding to the CV technology. In this study, specific CNN models were used to verify the appropriateness of this idea. Figure 2 is UAV-assisted bridge inspection to which this paper applies.

Most computer vision techniques use a CNN network as their backbone, with other specific algorithms that implement the CV functions. The CNN consists of an input layer, an output layer, and hidden layers in the middle of the network. Each hidden layer includes the convolution layer, the activation layer, the pooling layer, and the fully connected layer. Figure 3 shows a sample of overall CNN architecture.

The hidden layer performs the convolution operations to extract the feature information [23]. The pooling layer is placed after the convolution layer, and is also known as the subsampling layer; it reduces the width and height of neurons while retaining the depth to prevent overfitting during model training and to ensure the significant features of input data are preserved. The most commonly used pooling methods are average pooling and maximum pooling. The fully connected layer is located behind the convolution and pooling layers, and takes the role of a classifier in the overall convolution neural network. Due to the large number of parameters in the fully connected layer, some networks, such as ResNet and GoogLeNet, use a global average pooling layer instead of a fully connected layer to integrate the learned depth features. Finally, a loss function such as Softmax is used as the network objective function to make the final judgment.

The activation layer is combined with the pooling and convolution layers to solve non-linear problems during training. The commonly used activation functions are ReLU, Sigmoid, and Tanh. The normalization layer, also known as the Softmax function, is the last layer, through which the results of the network are output and the classifications of data are predicted. Specifically, in this layer, a normalized exponential function converts the input data into the probability that each sample belongs to a certain class.

## 3. CNN Model Verification for Bridge Image Classification

In this section, we studied the image classification task based on CV techniques, explored the applicability of the image classification task in determining the bridge type, and investigated the verification of a triple classification bridge image task using Resnet50 CNN network.

### 3.1. Resnet50 Network Structure

Resnet50 Network is the champion network of ILSVRC 2015, and was officially released in 2016 [24]. As a classical deep learning network, a CNN is better at image recognition than traditional methods. Deep network models such as AlexNet, VggNet, GoogleNet, and ResNet have been proposed and improved over several years. The recognition accuracy of the network model is improving, which brings up the issue of gradient disappearance and gradient explosion [25], and the residual network can solve these problems. At the same time, as a backbone network, it has the advantages of high robustness, multi-functionality, and ease of modification. The structure of Resnet50 is shown in Figure 4. Two residual blocks represent the IDBlock in Stage 1 to Stage 4 and do not change the dimension; the Conv Block consists of the residual blocks that add dimensions; each residual block contains three volumes.

### 3.2. Database Preparation

For the bridge type dataset, we collected 1200 images through web crawling and filtering, consisting of three different types of bridges: arched bridges, cable-stayed bridges, and suspension bridges. As a small-scale feature extraction task, Resize software was used to compress the image resolution and quality, and to standardize the image sizes to 256 × 256 pixels. Then, these images were divided into training, validation, and test sets, for model training, performance validation, and testing, respectively, see Table 1. Figure 5 shows some examples of bridge type dataset. 

### 3.3. Evaluation Metrics of Accuracy

In this study, the most popular performance metrics for image classification were used; namely, accuracy, precision, recall, F1-score, and AUC [26]. The following symbolic notations were used for ease of interpretation: *TP* (True Positive), *FP* (False Positive), *TN* (True Negative) and *FN* (False Negative); in this triple classification project, these were as assigned to the individual samples, and the classification results were judged separately for each category. According to the assigned notation, the accuracy, precision, recall, F1-score, and AUC, respectively, were calculated as follows:(1)$$\mathrm{Accuracy}=\frac{TP+TN}{NUM}$$
(2)$$\mathrm{Precision}=\frac{TP}{TP+FP}$$
(3)$$\mathrm{Recall}=\frac{TP}{TP+FN}$$
(4)$$F1-\mathrm{score}=\frac{2\mathrm{Precision}\times \mathrm{Recall}}{\mathrm{Precision}+\mathrm{Recall}}$$
(5)$$\mathrm{AUC}=P(P_{Ture}>P_{False})$$
where, accuracy represents the probability that the prediction is correct and *NUM* is the number of classified data sets, the sum of *TP*, *FN*, and *FP*. Precision describes the percentage of detected objects that genuinely fall into this class. In order for a project model to be used in practice, accuracy and precision are generally required to be higher than 80%. Recall describes the percentage of correctly detected objects compared to the total number of such objects. The F1-score is the harmonic average of model precision and recall, used to evaluate the overall function of the classifier. AUC (area under curve) means the area under the ROC curve. Simply, the trained classifier is used to predict a pair of positive and negative samples, and the probability of predicting positive samples is greater than that of negative ones.

### 3.4. Training and Validating

A training process was performed to optimize the initial parameters of the CNN and hyperparameter configurations, and to verify their accuracy by tracking the training performance over time. All work described was conducted with PaddlePaddle 2.1 frameworks for Windows and performed on a workstation configured with a graphics processing unit (GPU) (CPU. Intel^®^ Xeon^®^ W-2123, RAM: 32 GB, GPU: NVIDIA GeForce GTX 2080 8GB). In terms of the hyperparameters, the maximum training period was 80 epochs, and the images were resized to 224 × 224 for inputting. Due to the memory performance of the GPU, the maximum batch size was set to 10 in this task. Adam was adopted according to the optimization method of the loss function. Learning rate of 0.00625 was adopted, learning rate decay strategy is set to make the training loss values easy to converge. Learning rate could be adjusted according to the training results.

The training results were obtained. Figure 6a shows the training process, where the Y-axis is the training loss, and the X-axis is the number of training steps. The loss is the value that the neural network tries to minimize, and this is how the neural network learns, i.e., by adjusting the weights and biases to reduce the loss. After about 6000 training sessions, minimum values of 0.002922 for the training set and 0.1288 for the validation set eventually converged. The training curve of the validation set displayed more oscillations, and was harder to converge, than that of the training set, which was related to the fact that the validation set was trained less often. The loss curve of the training set did not overlap with the loss curve of the validation set, and the loss of the training set was smaller than the loss of the validation set, which provided an important basis for the judgement that the model was not overfitted. Figure 6c represents the rising curve of the accuracy of the training set. It can be observed that there is a corresponding relationship between the decrease of the loss value and the increase of the accuracy rate during training. As the training proceeded, the accuracy of the training set reached a maximum of 0.9687. To select the model with the best performance under the set hyperparameters, we used a model interval preservation strategy. In this experiment, the model parameters were saved every ten epochs; each model was used for validation; and the results were compared based on accuracy. As shown in Figure 6c, the highest accuracy of the validation set appeared in the model’s epoch 60, and reached 0.9167. Therefore, epoch 60 was chosen for further evaluation. To comprehensively evaluate the performance of the model, the precision and recall metrics were obtained; see Table 2 and Table 3. The three comprehensive evaluation metrics were higher than 96%, and the precision and recall values were close. The best classification evaluation indexes were for the arched bridges, the indexes for the cable-stayed bridges were slightly higher than the indexes for the suspension bridges, which were the lowest, but its F1-score indexes still reached more than 95%.

### 3.5. Testing Trained and Validated Resnet50

The overall average for bridge classification was 96.3%, the recall rate was 95.9%, and the F1 score reached 96.1%. Meanwhile, it was found that the recognition rate for arched and cable-stayed bridges was slightly higher than that for suspension bridges. Various types of samples were also tested, see Figure 7. In the recognition results for arched bridges, images with clear arched structures obtained better accuracy, and the more arched the structures and the simpler the background, the better the recognition effect. Misjudgment existed for some specific angles for arch bridges where the features of the subject were unclear. In the recognition results for cable-stayed bridges, the presence of a clear, ray-like image of a cable-stayed bridge in the image body achieved a high recognition effect, but the presence of adjacent bridges or towers in the image readily caused misjudgment. In the recognition results for suspension bridges, parallel cable-stayed structures were learned, and images with many parallel structures achieved a high recognition rate. In contrast, images with large cable-stayed towers at certain angles were recognized as either cable-stayed bridges or arched bridges. These data indicate that image data should be not acquired from inappropriate angles in order to prevent unclear subjects.

## 4. CNN Model Verification for Bridge Part Detection

In this section, we studied the object detection task based on CV techniques, explored the applicability of the target detection task in determining bridge sections, and investigated the validation of the bridge section detection task using the YOLOv3 model based on the CNN network.

### 4.1. YOLOv3 Model Structure

YOLOv3 is one of the object detection models, with good performance in terms of both detection speed and accuracy, and it is considered to have high accuracy for small targets [27]. The structure of the YOLO v3 model is shown in Figure 8. YOLO v3 introduced the residual calculation method based on YOLOv1 and v2, deepening the network layers. The backbone network has 53 convolutional layers, called Darknet-53. It mainly consists of convolutional layers and residual connected blocks, with convolution kernels of 3 × 3 and 1 × 1. The detection network of YOLO v3 introduced the concept of multiscale prediction. The number of detection layers was increased from 1 to 3, and the three layers correspond to 13 × 13, 26 × 26, and 52 × 52 feature maps, respectively. More details can be found in the literature [27].

### 4.2. Database Preparation

Through internet crawling and filtering, we collected 300 images of different types of bridge main girders and piers as the dataset for bridge parts. Resize software was used to compress the image resolution and quality, and to normalize the image sizes to 256 × 256 pixels. These images were labeled with bounding boxes for the girders and piers using the LabelMe program, with a total of 1686 targets, and then divided into training, validation, and test sets for model training, verification, and testing, respectively, as shown in Table 4. For training, we used an image argumentation technique to expand the training sample by a factor of 4 to obtain better model performance and robustness. Figure 9 show some examples from the bridge types dataset.

### 4.3. Evaluation Metrics

In the object detection task, intersection over union (IoU) was used to determine the positivity of the samples. This is a metric to evaluate the correlation between the detected frames and the actual frames based on using the overlap ratio to determine the positivity or negativity of the prediction, and it is defined as shown in Figure 10. In this study, prediction results with an IoU greater than 0.5 were set as positive, and then the indexes *TP, FP, FN, PN* were obtained. The detection metrics, AP and mAP, were introduced in addition to precision and recall due to the complexity of the object detection task in deriving both the types and the bounding boxes. AP is the cumulative integral of precision over the recall, and it evaluates the detection effectiveness for a class of objects, and mAP is the mathematical expectation of the AP values for each class. In this project, k = 2. AP and mAP were calculated as follows:(6)$$AP=\int_{0}^{1}pdr$$
(7)$$mAP=\frac{\sum_{i=1}^{k}AP_{i}}{k}$$

### 4.4. Training and Validating

The training platform and hardware used in this task were consistent with that used in the bridge image classification task. In terms of hyperparameters, this experiment performed best using an ImageNet pre-training model with a batch size of 6, and 270 epochs. A dynamic learning rate strategy was used to warm-up the learning rate at the beginning of the training session to prevent gradient explosion, and the learning rate was reduced toward the end of the training period to ensure the model readily converged. Specifically, the learning rate was increased linearly from 0 to 0.00125 in the first 2000 steps, and reduced over the last 1500 steps until the end of the training.

Figure 11a shows the training process for the model. After about 9000 training steps, the minimum value of the model was 2.224, and after reaching this minimum value, it oscillated and converged around that minimum value. It can be observed that the minimum loss value occurred in the region where the learning rate was decreasing. The loss value of the model did not then decrease as the learning rate decreased further, rather it converged in the oscillation, which was considered to show effective training [21]. In this study, the loss value of training during the dynamic learning rate was reduced by 13.67 compared to that of the fixed learning rate. 

This task also used an interval saving strategy, where the model parameters were saved every 30 epochs, and each model was used for validation. The results were compared according to mAP in order to select the optimal model. We derived the comprehensive model evaluation metrics, as shown in Table 5. The identification accuracy of the main girders was 93.5%, and the recall rate was 72.5%. The accuracy for bridge piers was 82.6%, the recall rate was only 54.9%, and the overall mAP value was 75.5.

### 4.5. Testing Trained and Validated YOLOv3

The top of Figure 12 shows several thresholds which were set and compared in the test results. Threshold is considered an important issue; all bounding boxes with confidence levels less than the threshold were filtered out. In this case, girder detection had a high performance, with a confidence level of 0.93, which was displayed regardless of the threshold value selected. Conversely, the confidence for all types of piers was lower than 0.6; when a threshold value higher than 0.6 was selected, the results below this value were not displayed. Among these, single large volume piers were most easily detected, followed by columnar piers, and finally small volume piers. This was attributed to the proximities and the type of targets in the image data, which influenced the priority of detection. Several test results are shown below the images, including a comparison of the truth targets with the test targets. The model was considered to have high performance for the detection of main girders, which could be detected at high thresholds, probably because the objects are large in the image data, and the features are easily distinguished from the background. However, there were also problems with missing large girders in small targets, and inaccurate bounding boxes, which were thought to be caused by insufficient training data and the data itself not being accurately labeled. Lowering the detection threshold facilitated the detection of bridge piers. The model was more effective in determining piers without overlap, however, there were still faults in the detection of small targets. For piers with overlap, the overlapping part rendered the pier boundaries unclear and easily confused with the background, thus causing the problem of missed detection. Therefore, it was very important for the UAV inspection process to select the correct angle and distance to avoid overlapping parts. In addition, a problem related to the optimization of the overlapping area detection algorithm needs to be further studied.

## 5. CNN Model Verification for Bridge Component Segmentation

In this section, we investigated the instance segmentation based on CV techniques, explored the applicability of the instance segmentation task in identifying and segmenting small bridge components, and validated the bridge component segmentation task using the Resnet50-based Mask-RCNN model.

### 5.1. Mask RCNN Model Structure

Mask-RCNN [16] is one of the advanced image segmentation CNN models. This variant of the deep neural network can detect objects in images and generate high-quality segmentation masks for each instance. It is the best paper of ICCV2017, achieving both object detection and instance segmentation in a single network. The algorithm runs at almost 5 fps on a single GPU and outperforms all three challenges of the COCO dataset: instance segmentation, bounding-box object detection, and person–keypoint detection. Mask R-CNN parallels the existing classification and bounding box regression branches by adding a branch of the predictive segmentation mask to each region of interest (ROI), as shown in Figure 13. The mask branch is a small FCN applied to each ROI to predict the segment mask in each pixel. Moreover, the mask branch only adds a small amount of computing overhead, thus realizing a fast system [16,28]. The identification and segmentation of bridge components is essential for automatic bridge inspection. There are many types of components, and the clear identification and analysis of the status of certain components is an important part of bridge inspection.

### 5.2. Database Preparation

In image recognition, the task of collecting detailed images is considered challenging. High-quality, massive images can greatly improve the effectiveness of model training. In this experiment, we attempted to use manual inspection record images to train CNN models. These images were filtered from the bridge inspection records of Yokohama City, Japan, of the past five years, and were captured manually. These images were detailed images of bridge components, such as bolted joints, girders, bearings, etc. We filtered the images to obtain the desired bearing and girder images. We selected 843 images of bridge bearing and main girder members, the image size was standardized to 256 × 256, and then these images were annotated. We used LabelMe to perform the labeling work, and the instance segmentation task required the labeling of the target pixels to generate a mask. The labeled data were organized to generate the dataset, as shown in Table 6. Figure 14 shows some examples from the bridge component dataset.

### 5.3. Training and Validating

Training was carried out with a batch size of 6, and 60 epochs. A dynamic learning rate strategy was also adopted in this process. The learning rate was 0.0025, the initial learning rate was set to 0.00041, the number of warm-up steps was 500, and the learning rate decayed at 55 and 58 epochs of training to facilitate model convergence. Figure 15 shows the process of Mask RCNN training. As the training proceeded, the backpropagation algorithm came into play. The loss function value decreased rapidly at first and then oscillated slowly downward as the training proceeded, converging to a minimum value of 0.03473 at 7200 steps. An interval preservation strategy was also adopted to select the best model; the results were compared based on mAP. Since object identification and segmentation are computed in parallel in Mask-RCNN, the mAP values of these two branches were verified simultaneously. The best result of the validation set appeared in epoch 54 of the model, where the bounding box and mask mAPs reached 53.99 and 50.48, respectively. Thus, epoch 54 was used for further validation. Table 7 and Table 8 show the evaluation metrics of the model. The valuation metrics used in the instance segmentation were the same as those used in the object detection. It was found that the object box recognition of bearings had the same evaluation metrics as the mask recognition, with precision and recall of 95.45, and an AP value of 95.08%. In the segmentation of the main girder, the accuracy of the bounding box was 87.8%, while the precision of the mask was 82.2%. Notably, the index value of the girder mask was the lowest among all the data. The lower accuracy of the girders compared to the bearings was considered to be related to the complexity of the image backgrounds as there are also bridge components with characteristics similar to those of the main girders in the girder images, such as crossbeams, diagonal braces, and stiffeners, which may have affected the identification of the main girders.

### 5.4. Testing Trained Mask-RCNN

Instance segmentation is different from general object detection and semantic segmentation, in that it recognizes and segments targets while distinguishing between the same targets. In detecting bridge components, there are usually multiple parts of the same component. In this experiment, we developed a visualization module for component recognition based on instance segmentation to effectively identify different types of components. Figure 16 shows the test results of the bridge component instance segmentation. In the visualization results, the type of the component, the confidence value, the object bounding box, and the object mask are the outputs. It can be seen that the visualization distinguished the different color bridge components in both single- and multi-object recognition and segmentation. In the bearing segmentation images, the background around the bearing was purer, the main body of the recognition was prominent, the bearing and the subsidiary structure could be segmented accurately, and the single bearing recognition was stronger than multiple bearing recognition, and both had very high confidence level. In comparison, the complexity of girder identification was much higher, because the areas where the main beams were located had many other bridge components, such as pipes, stiffeners, and bolt plates. The detection of a single girder necessitated images to be taken at a closer distance and with less background to obtain the best effect, and scenes where multiple girders were present had better robustness against the interference of pipes or corrugated webs, and so a clear boundary could be segmented. For truss beams, although multiple beams could be detected, there were cases of missed detection and incomplete edge segmentation. This point was attributed to the fact that the truss beam is more complex, with many components, and the features of the bolt plate were not fully learned. In general, the model was considered to have good detection and segmentation capability for both girder and bearing components.

## 6. Conclusions

In this study, we deconstructed the tasks of automatic bridge image recognition and validated these tasks using computer vision models. The results showed that computer vision techniques are highly applicable to recognition tasks involving bridge types, large parts, and small components. In particular, the Resnet50 network used in the study achieved 96.3% accuracy in the bridge type classification task, the YOLOv3 model achieved an overall mAP of 0.7557 in the bridge object recognition task, and the Mask-RCNN model achieved an overall mAP of 0.87 in the small part segmentation task. More bridge types and components are expected to be included in future studies. The optimization of the dataset and the image processing methodology requires further work. In addition, in future, computer vision models should be investigated and improved, to obtain better model performance.

## Figures and Tables

**Figure 1 sensors-22-03502-f001:**
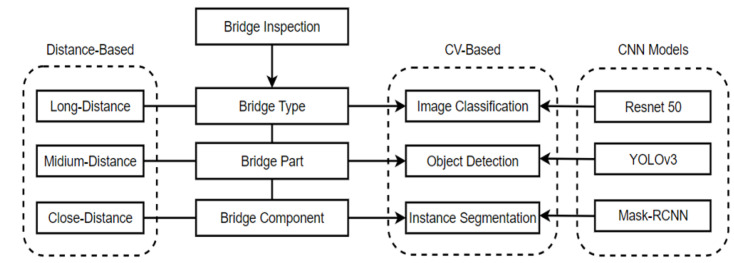
The overall structure of multi-dimensional bridge inspection.

**Figure 2 sensors-22-03502-f002:**
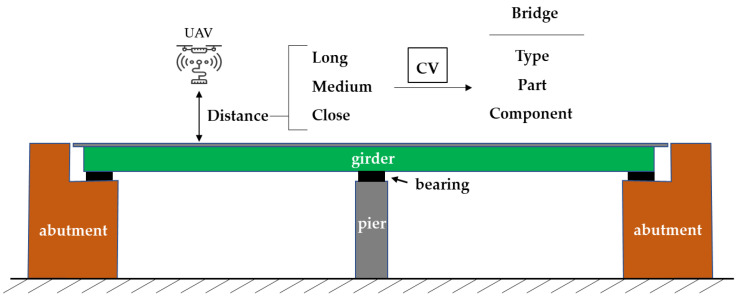
UAV-assisted bridge inspection to which this paper applies.

**Figure 3 sensors-22-03502-f003:**
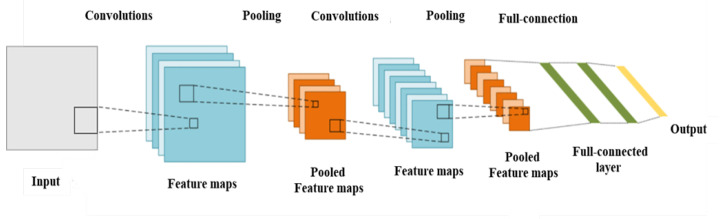
Sample overall CNN architecture.

**Figure 4 sensors-22-03502-f004:**
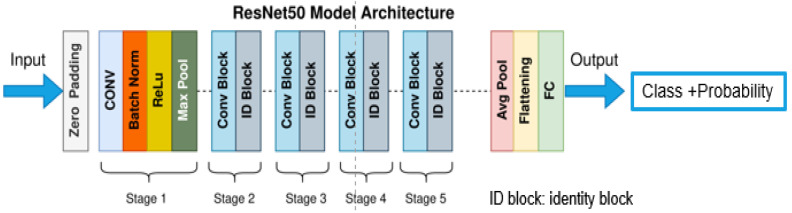
Resnet50 network architecture.

**Figure 5 sensors-22-03502-f005:**
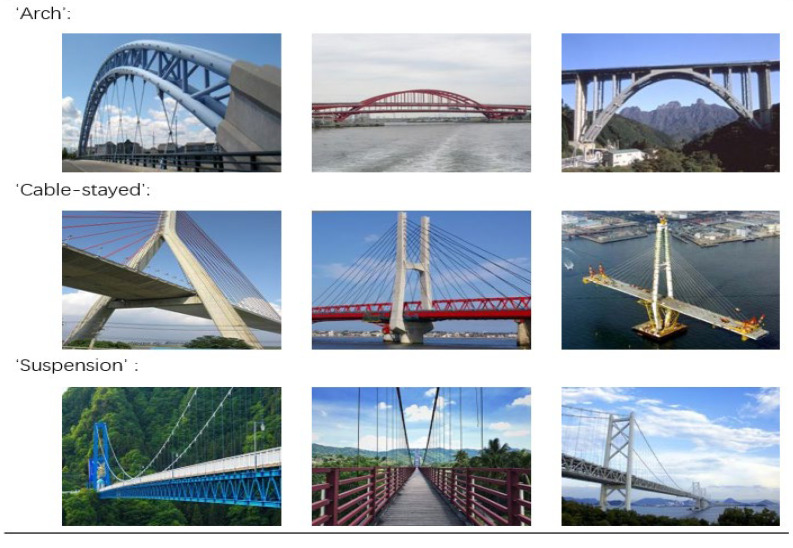
Example of bridge type dataset.

**Figure 6 sensors-22-03502-f006:**
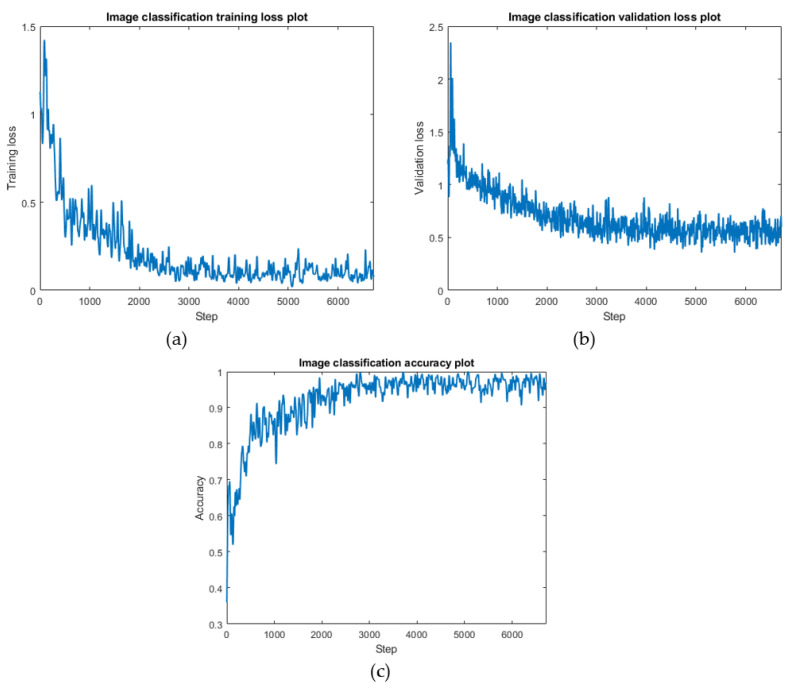
Model training performance (**a**): Training set loss variation diagram. (**b**): Validation set loss variation diagram. (**c**): Training Accuracy.

**Figure 7 sensors-22-03502-f007:**
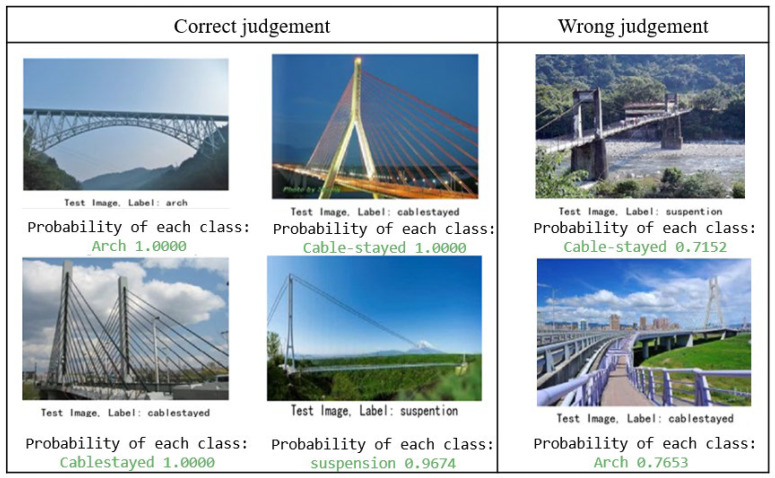
Example prediction results for bridge types.

**Figure 8 sensors-22-03502-f008:**
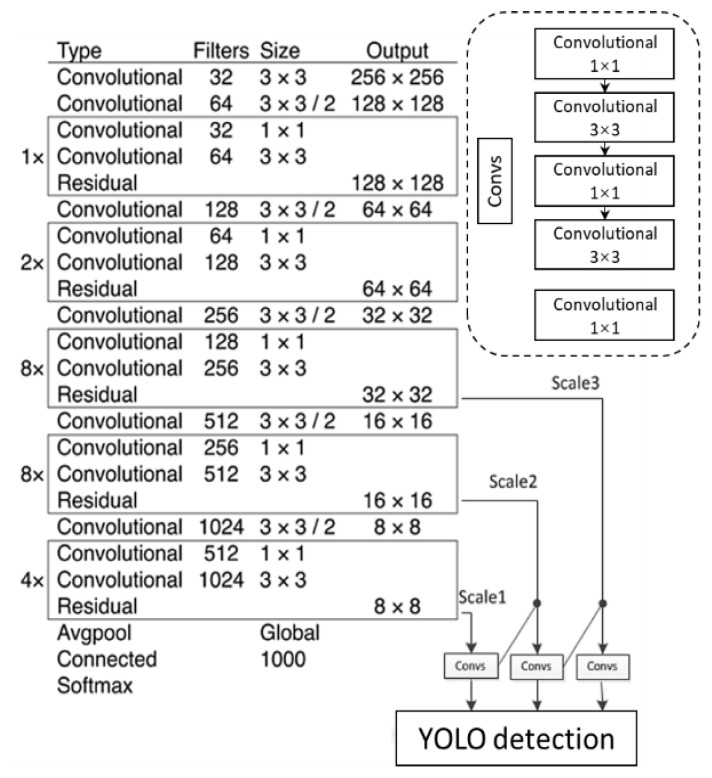
YOLO v3 backbone network and model structure.

**Figure 9 sensors-22-03502-f009:**
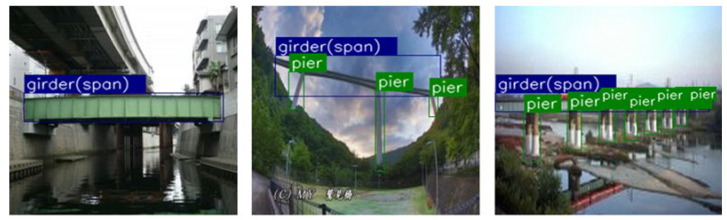
Examples from the bridge types dataset.

**Figure 10 sensors-22-03502-f010:**
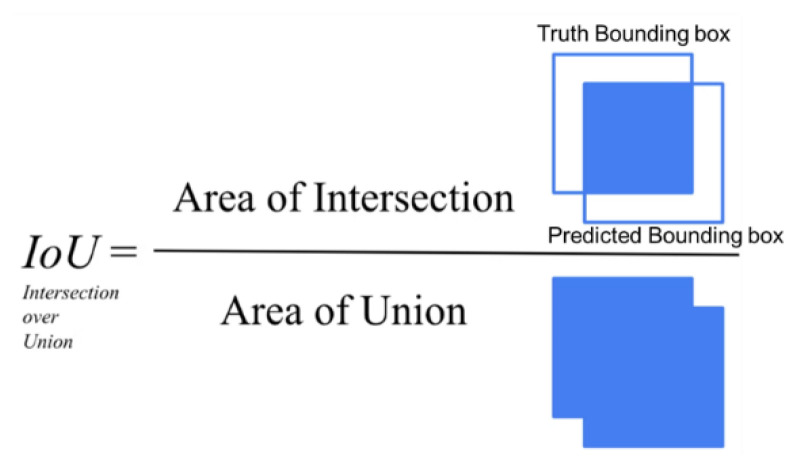
Definition and Calculation of IoU.

**Figure 11 sensors-22-03502-f011:**
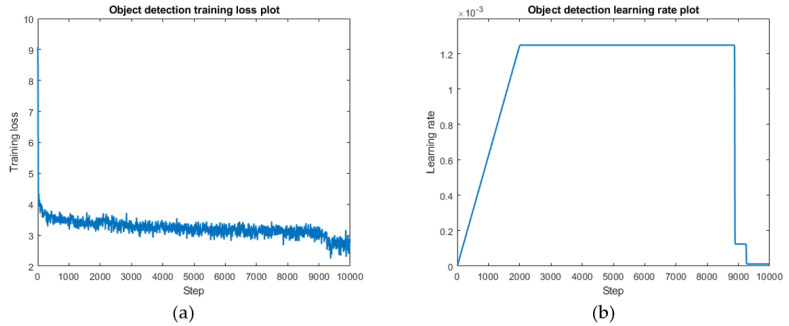
Model training performance (**a**): Training loss curve. (**b**): Dynamic learning rate strategy.

**Figure 12 sensors-22-03502-f012:**
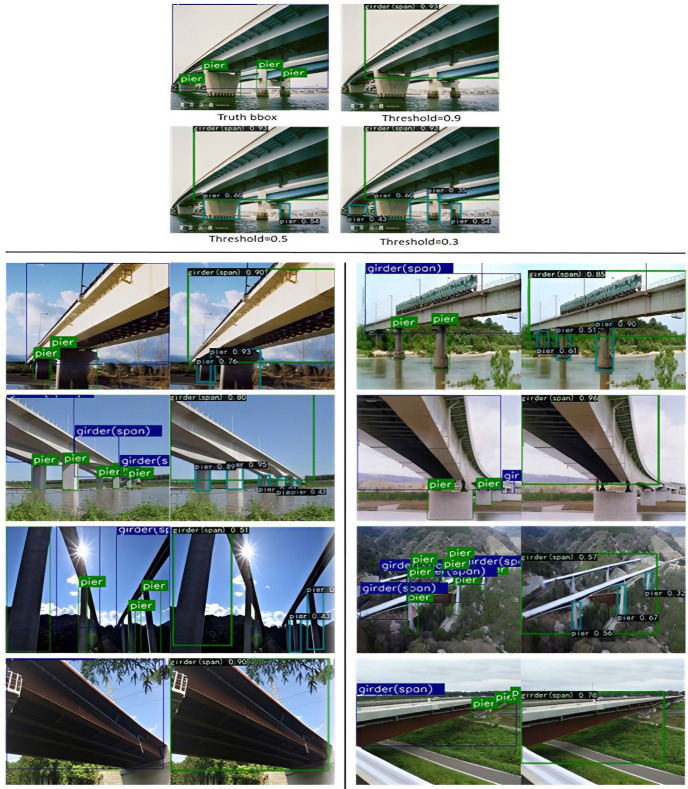
Samples of bridge part detection results.

**Figure 13 sensors-22-03502-f013:**
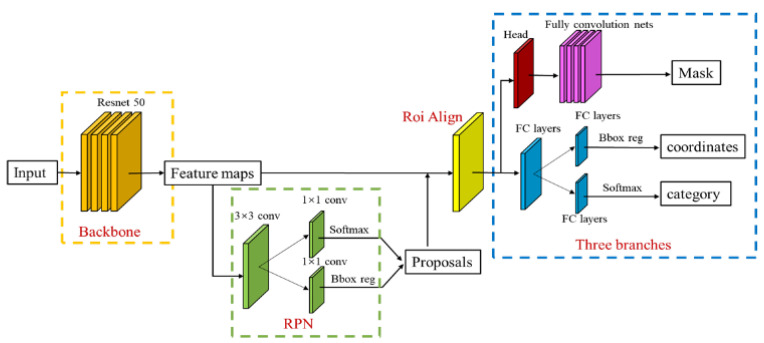
Mask-RCNN Model Architecture.

**Figure 14 sensors-22-03502-f014:**
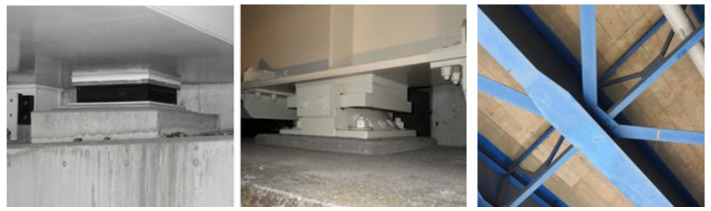
Examples from the bridge component dataset.

**Figure 15 sensors-22-03502-f015:**
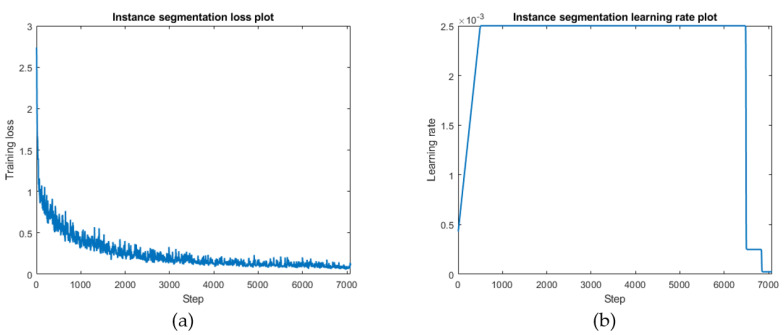
Model training performance (**a**): Training loss curve. (**b**): Dynamic learning rate strategy.

**Figure 16 sensors-22-03502-f016:**
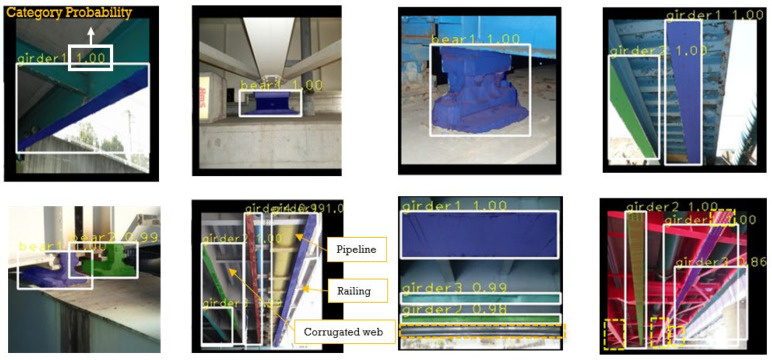
Samples from the bridge component detection results.

**Table 1 sensors-22-03502-t001:** The bridge type database.

Class	Total	Training (70%)	Validation (20%)	Testing (10%)
All (3)	1200	840	240	120
Arch	400	275	84	41
Cable-Stayed	400	279	77	44
Suspension	400	286	79	35

**Table 2 sensors-22-03502-t002:** Overall validation results.

Class	Precision	Recall	F1-Score
Overall	96.29%	96.25%	96.25%

**Table 3 sensors-22-03502-t003:** Classified validation results.

Class	Precision	Recall	F1-Score	AUC
Arch	97.62%	97.62%	97.62%	99.86%
Cable-Stayed	97.30%	93.51%	95.36%	99.63%
Suspension	93.90%	97.47%	95.65%	99.57%

**Table 4 sensors-22-03502-t004:** The bridge parts database.

Class	Total	Training (75%)	Validation (20%)	Testing (20%)	Num of Object
All	300	225	60	15	1686
Girder	295	221	59	15	576
Pier	222	172	40	10	1110

**Table 5 sensors-22-03502-t005:** Overall validation results.

Class	Precision	Recall	AP
Girder(span)	93.55%	72.50%	79.51%
Pier	82.64%	54.95%	71.62%
		Overall mAP	75.57%

**Table 6 sensors-22-03502-t006:** The bridge parts database.

Class	Total	Training (70%)	Validation (20%)	Testing (10%)
All	843	591	168	84
Bearing	442	305	86	51
Girder	401	286	82	33

**Table 7 sensors-22-03502-t007:** Validation results for the bounding box.

Class	Precision	Recall	AP
Bearing	95.45%	95.45%	95.08%
Girder	87.85%	85.45%	86.51%
		Overall mAP	90.80%

**Table 8 sensors-22-03502-t008:** Validation results for the mask.

Class	Precision	Recall	AP
Bearing	95.45%	95.45%	95.08%
Girder	82.24%	80.00%	79.26%
		Overall mAP	87.17%

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
