# Peer review of "Multilevel Structural Components Detection and Segmentation toward Computer Vision-Based Bridge Inspection"

_sensors, 2022, doi:10.3390/s22093502_

Round 1

Reviewer 1 Report

The paper presents the research on an innovative approach of Structural Health Monitoring. As the employing robotic inspection tools as UAVs is still in phase of promising tests, presented approach of visual data acquired by specialized drones and their analysis with use of CV technology and verification with CNN models, certainly shall be considered an important attempt. The paper addresses a certain gap in this field.

The research part is well-preceded by the Introduction based on exhaustive, very interesting and well-structured state-of-art report. Some of the elements as recalling OCR or face recognition could be considered as irrelevant to the text, still, I assume it is worth to show a bit wider picture as Authors proposed.

The research supported with well-derived analytical part, itself is well-designed and executed, the methodology is carefully and clearly explained, the results are well described and concluded.

Authors do draw a line towards new research directions which is worth to mention.

I have not encountered any issues worth correcting, neither formal, editing and typo errors.

The only remark which could be formulated is that the authorship of Figures 1, 2, 3, 10 is not clear.

The paper is well-written in English, draws and tables are clear to the reader. The paper is edited on a high level.

Author Response

Thank you very much for your appreciation and encouragement, which makes us feel the value of our research and motivates us to continue our efforts.

I will introduce the revised points below.

_______________________________________________________

"""

Some of the elements as recalling OCR or face recognition could be considered as irrelevant to the text, still, I assume it is worth to show a bit wider picture as Authors proposed.

""

↑This point has been updated in the paper and awider view of the application of the technology has been added.

""

The only remark which could be formulated is that the authorship of Figures 1, 2, 3, 10 is not clear.

""

↑Figure 1 is the original.
In Fig. 2 and Fig. 10, we refer to various CNN network structure patterns on the web and create our own.
Figure 3 is based on Wikipedia's introductory material.

_____________________________________________________

Thank you sincerely for your comments, which will help us to grasp the significance of academic  and make our research and submission process more rigorous.

Yours sincerely

YU WEILEI

Reviewer 2 Report

Dear Authors, 
congratulations for the work done.
The paper aims at proposing a framework for the multilevel inspection of bridge based on computer vision (CV) technology and verification using Convolutional Neural Network (CNN) models. A data set of thousands of brige-related images, which are classified based on the distance (long, medium, short), were used in this study.
Results show (1) a classification accuracy of 96.29% for the model used to recognize bridge type (long-distance database), (2) a classification accuracy in the range 82.64%-93.55% for the model used to recognize bridge components (medium-distance database), and (3) a classification accuracy in the range 87.17%-90.80% for the model used to carry out segmentation and recognition of bridge components (short-distance database).

In general, the paper is interesting and well written. The following comments have been provided to improve the readability of the paper:
1. Abstract: Please, define all the acromyms into this section (e.g., UAV, CNN, etc.).
2. Lines 49-57: This part needs at least a couple of references that support your statements and that refer to infrastructure monitoring using artificial intelligence (see e.g., 10.1016/j.autcon.2018.10.019; 10.3390/a13040081). 
3. Lines 84-94: These statements need at least a reference (please, do the same for the statement at lines 212-213).
4. Line 208: A figure that shows examples of each type of bridge that you took into account could be added, please. The same thing can be done for the medium-distance and short-distance dataset.
5. Equations 2-6: Please, define all the symbols used in these equations. It could be interesting to add explanations related to these performance metrics (e.g., maximum and mimimum expected/typical values, or typical trends). 
6. Figure 8-a: The value "5.915" is difficult to be identified in the figure. This can make the reader confused.
7. Figure 9: If possible, please improve the quality of this figure.

Best regards.

Author Response

Thank you very much for your appreciation and encouragement, which makes us feel the value of our research and motivates us to continue our efforts.

I will introduce the revised points below.

_____________________________________________________________________________________

Abstract: Please, define all the acromyms into this section (e.g., UAV, CNN, etc.).

# Thank you for your comment, this point has been updated in the manuscript

Lines 49-57: This part needs at least a couple of references that support your statements and that refer to infrastructure monitoring using artificial intelligence (see e.g., 10.1016/j.autcon.2018.10.019; 10.3390/a13040081).

# Thank you for your comment, this point has been updated in the manuscript, I have read the two pieces of literature you provided and have cited them as references..

Lines 84-94: These statements need at least a reference (please, do the same for the statement at lines 212-213).

# Thank you for your comment, this point has been updated in the manuscript (Lines 84-94, lines 212-213)

Line 208: A figure that shows examples of each type of bridge that you took into account could be added, please. The same thing can be done for the medium-distance and short-distance dataset.

# Thank you for your comment, have added bridge sample images into manuscript.

Equations 2-6: Please, define all the symbols used in these equations. It could be interesting to add explanations related to these performance metrics (e.g., maximum and minimum expected/typical values, or typical trends).

# Thank you for your comment, we added the definition of symbols and explained it .

Figure 8-a: The value "5.915" is difficult to be identified in the figure. This can make the reader confused.

# Here is a writing error, thank you for pointing it out. The real value is 2.224. This was revised in the manuscript

Figure 9: If possible, please improve the quality of this figure.

# Thank you for your comment, we update this figure, now it's clear.

_________________________________________________________________________________

Thank you sincerely for your comments, which will help us to grasp the significance of academic and make our research and submission process more rigorous.

Best regards,

Weilei YU